# Poly(ADP)-Ribosylation Inhibition: A Promising Approach for Clear Cell Renal Cell Carcinoma Therapy

**DOI:** 10.3390/cancers13194973

**Published:** 2021-10-03

**Authors:** Yaroslava Karpova, Danping Guo, Peter Makhov, Adam M. Haines, Dmitriy A. Markov, Vladimir Kolenko, Alexei V. Tulin

**Affiliations:** 1Department of Biomedical Sciences, School of Medicine and Health Sciences, University of North Dakota, Grand Forks, ND 58202, USA; iaroslava.karpova@und.edu (Y.K.); danping.guo@und.edu (D.G.); 2Koltzov Institute of Developmental Biology of Russian Academy of Sciences, 119334 Moscow, Russia; 3Fox Chase Cancer Center, Philadelphia, PA 19111, USA; Petr.Makhov@fccc.edu (P.M.); vladimir.kolenko@fccc.edu (V.K.); 4Department of Cell Biology and Neuroscience, School of Osteopathic Medicine, Rowan University, Stratford, NJ 08084, USA; adam.haines@jefferson.edu (A.M.H.); markovdm@rowan.edu (D.A.M.)

**Keywords:** PARylation, PARG, PARP-1 inhibitors, poly(ADP-ribose), RCC, cancer cell

## Abstract

**Simple Summary:**

We have shown that Clear Cell Renal Cell Carcinoma (ccRCC) is prone to accumulate poly(ADPribose) due to increase in PARP-1 and decrease in PARG enzymes. Reduction of pADPr by PARP-1 inhibition or PARG overexpression disrupts renal carcinoma cell malignancy. We have shown that ccRCC is prone to accumulate poly(ADPribose) due to increase in PARP-1 and decrease in PARG enzymes. Reduction of pADPr by PARP-1 inhibition or PARG overexpression disrupts renal carcinoma cell malignancy. Transcriptome analysis linked observed phenotype with changes in gene expression levels for lipid metabolism, interferon signaling, and angiogenesis pathways along with the changes in expression of key cancer-related genes. While apoptosis level didn’t change under pADPr reduction with chemical or genetic approaches, our work supports the idea that gene expression modulation is a primary cause for observed anti-tumor effects.

**Abstract:**

Poly(ADP-ribose) polymerase 1 (PARP-1) and glycohydrolase (PARG) enzymes regulate chromatin structure, transcription activation, and DNA repair by modulating poly(ADP-ribose) (pADPr) level. Interest in PARP-1 inhibitors has soared recently with the recognition of their antitumor efficacy. We have shown that the development of clear cell renal cell carcinoma (ccRCC) is associated with extreme accumulation of pADPr caused by the enhanced expression of PARP-1 and decreased PARG levels. The most severe misregulation of pADPr turnover is found in ccRCC specimens from metastatic lesions. Both, classical NAD-like and non-NAD-like PARP-1 inhibitors reduced viability and clonogenic potential of ccRCC cell lines and suppressed growth of ccRCC xenograft tumors. However, classical NAD-like PARP-1 inhibitors affected viability of normal kidney epithelial cells at high concentrations, while novel non-NAD-like PARP-1 inhibitors exhibited activity against malignant cells only. We have also utilized different approaches to reduce the pADPr level in ccRCC cells by stably overexpressing PARG and demonstrated the prominent antitumor effect of this “back-to-normal” intervention. We also generated ccRCC cell lines with stable overexpression of PARG under doxycycline induction. This genetic approach demonstrated significantly affected malignancy of ccRCC cells. Transcriptome analysis linked observed phenotype with changes in gene expression levels for lipid metabolism, interferon signaling, and angiogenesis pathways along with the changes in expression of key cancer-related genes.

## 1. Introduction

Renal cell carcinoma (RCC) is the most common form of kidney cancer with limited therapeutic options for metastatic disease. According to the 2004 WHO classification, several histological RCC subtypes are recognized with clear cell RCC being the most frequent (ccRCC; 70–75% of cases) [1]. The vast majority of sporadic ccRCC cases are associated with loss of von Hippel-Lindau tumor suppressor gene (*VHL*) function [2,3]. To date, the best-characterized targets of VHL protein (pVHL) are the α-subunits of the hypoxia-inducible transcription factors HIF-1α and HIF-2α [4,5,6]. Oxygen-dependent hydroxylation of HIFs at specific proline residues by prolyl hydroxylase enzymes triggers binding of pVHL, ubiquitination, and subsequent proteasomal degradation [6,7,8]. Loss of function of pVHL in cells results in the stabilization of HIF-1α and HIF-2α and the enhancement of their transcriptional activity, leading to constitutive activation of hypoxia pathways, dramatic metabolic changes, and malignant transformation [4,5,6].

Poly(ADP-ribose) polymerase 1 (PARP-1) plays a critical role in preserving genomic integrity, transcriptional regulation, and cell fate determination [9]. PARP-1 utilizes NAD to synthesize poly(ADP-ribose) (pADPr), resulting in both auto modification and the modification of acceptor proteins. Poly(ADP-ribose) glycohydrolase (PARG) removes pADPr subunits from proteins, thus reversing the action of PARP enzymes [10]. PARP-1 and PARG expression and activity are frequently dysregulated in tumors of various origins, which affects endogenous pADPr levels [11,12,13,14,15].

In this study, we investigated the expression levels of pADPr in ccRCC and normal kidney tissue specimens. The most severe misregulation of pADPr turnover was found in ccRCC specimens obtained from metastatic lesions. To address this issue, we utilized different approaches to reduce the pADPr level in ccRCC cells and demonstrated the antitumor effect of these “back-to-normal” interventions. We tested classical PARPs inhibitors, novel non-NAD-like PARP-1 inhibitors, and PARG overexpression system in several established and patient-derived ccRCC cell lines. We followed transcriptome changes upon pADPr reduction and linked such changes to the observed malignancy-affected phenotype.

## 2. Materials and Methods

### 2.1. Materials

Olaparib (HY-10162), rucaparib (HY-10617A), and sunitinib (HY-10255A) were purchased from MedChemExpress. 5F02 was obtained from the ChemDiv Representative Diversity Set, comprising 50,000 molecules. Samples were validated by ChemDiv1H-NMR and HPLC/LCMS for ≥95% purity. For cell culture studies, inhibitors were dissolved in DMSO at 10 mM stock concentration and kept at −80 °C protected from light. Final dilutions were made in culture media. For the in vivo studies, drug compaunds were diluted from powder in 10% 2-hydroxypropyl-β-cyclodextrin on PBS and used freshly made for each injection. Doxycycline (Millipore-Sigma, Darmstadt, Germany, #9891) and puromycin (Clontech/Takara, Mountain View, CA, USA, 631,306) were dissolved in DMSO at stock concetrations of 100 ug/mL and 1 mg/mL, respectilvely, and kept at −20 °C.

### 2.2. Cells and Culture Conditions

The 786-O human RCC cell line was obtained from ATCC (Rockville, MD, USA). The PNX0010 human ccRCC cell line, which was described previously [12], was a kind gift of Dr. Igor Astsaturov, MD, PhD. (Fox Chase Cancer Center, Philadelphia, PA, USA). The 769-P, HRC-20, HRC-24, HRC-31, HRC-45, HRC-51, HRC-78, SK-26b, and SK-45 human renal cell carcinoma cell lines were a kind gift of Dr. Joseph Testa (Fox Chase Cancer Center, Philadelphia, PA, USA). Normal kidney epithelial cell line NK677 was obtained from the Fox Chase Cancer Center Cell Culture Facility. Initial stocks were cryopreserved, and at every 6-month interval, a fresh aliquot of frozen cells was used for the experiments. Cells were cultured in RPMI-1640 (Bio-Whittaker, Walkersville, MD, USA) supplemented with 10% FCS (Hyclone, Logan, UT, USA), gentamicin (50 mg/L), sodium pyruvate (1 mM), and non-essential amino acids (0.1 mM) under conditions indicated in the figure legends.

Lenti-X 293T cells (Clontech/Takara, 632,180) were maintained in 90% DMEM (Millipore-Sigma, D5796) with 10% FBS, 100 units/mL penicillin G sodium, and 100 ug/mL streptomycin sulfate, and used as the host cell line for lentiviral packaging. HT1080 (ATCC Manassas, VA, USA CCL-121) cells for lentiviral titration were grown in EMEM (ATCC, 30-2003).

### 2.3. PARP-1 Inhibitory Assay in Human Cell Culture

Different concentrations of new non-NAD-like PARP-1 inhibitors or classical PARP-1 inhibitors were added to the cells cultured in complete medium. After 24 or 48 h, cells were lysed, and protein samples were analyzed with SDS-PAGE and Western blot using anti-pADPr antibody.

### 2.4. Cell Proliferation Assay

Normal cells NK677 normal kidney epithelial cells were plated at a density of 10^4^ cells/well (100 µL) in a 12-well plate. Cells were transiently transfected with PARG:GFP encoding plasmid using FuGENE (Promega, Madison, WI, USA) transfection reagent according manufacturer protocol. Alamar Blue Reagent was added, and fluorescence readings were taken after specified number of days.

### 2.5. Cell Survival Assay under PARP-1 Inhibitors Treatment

The assay was based on the published protocol [16]. Cells were plated into a 24-well plate at a density of 2000 cells/well. Cells were allowed to adhere overnight at 37 °C and were treated with increasing concentrations of non-NAD-like PARP-1 inhibitors or classical PARP-1 inhibitor olaparib for 14 days. Colonies were fixed with 70% ethanol for 10 min and stained with 0.25% methylene blue in 30% ethanol for 10 min. After that, staining solution was removed, and plates were rinsed with water. Colonies consisting of 50 cells, or more were counted. Data were fitted to exponential and logarithmic decay models using the nonlinear curve fitting module of Statistica 7.0 software. The best fitting models for each inhibitor are represented on the chart. Plating efficiencies (PE) were calculated as follows: PE = amount of colonies/number of cells seeded. The surviving fraction (SF) was calculated as follows: SF = amount of colonies/number of cells seeded × PE.

### 2.6. Measurement of Cell Viability under PARP-1 Inhibitors Treatment

Cell viability was analyzed by CellTiter Blue assay (Promega). Effective doses (ED) were calculated using XLfit, a Microsoft Excel version 16 add-in.

### 2.7. Assessment of In Vivo Tumor Growth

All animal procedures were done in accordance with the institutional guidelines on animal care and with appropriate institutional certification. For in vivo studies, 1 × 10^6^ PNX0010 cells were inoculated s.c. in the flank region of 6-week-old male C.B17/Icr-scid mice using a 27-gauge needle. Mice received an autoclaved 2018SX diet (Harlan Teklad, Madison, WI, USA) and sterile water. When tumors reached a mean volume of about 100 mm^3^, the mice were randomly assigned to the treatment and control groups (*n* = 7 mice per group). Animals were treated with the non-NAD-like inhibitor 5F02 (23 mg/kg, i.p.), classical PARP-1 inhibitor olaparib (Olap, 50 mg/kg, i.p.), multi-targeted tyrosine kinase inhibitor sunitinib (40 mg/kg, p.o.), or vehicle (10% 2-hydroxypropyl-β-cyclodextrin on PBS) 5 days a week. Tumor volumes were calculated as (volume = 0.52 × (width)^2^ × length). Data were fitted to exponential growth models using the nonlinear curve fitting module of Statistica 7.0 software. Error bars correspond to standard deviation.

### 2.8. Tumor Immunohistochemistry

Formaldehyde-fixed paraffin-embedded tissue sections were stained for pADPr by immunohistochemistry as described previously [17]. 

### 2.9. Apoptosis Detection in Tumors

We used the ApopTag^®^ Fluorescein In Situ Apoptosis Detection Kit (Millipore-Sigma, # S7110) to detect the occurrence of cell death. Tumors were fixed and sectioned the same as for immunohistochemistry. Tissues were processed according to the manufacturer’s recommendations.

### 2.10. Generation of Stable ccRCC Cell Lines with Human PARG Overexpression Lentivirus Construct under Doxycycline Inducible Promoter

pLVX-TetOne-Puro (Clontech/Takara, 631,847) with puromycin selection was used as the backbone vector for an inducible gene expression system. Human PARG cDNA were cloned to pLVX-TetOne-Puro, followed by P2A cleavage site and mClover3 gene that encodes fluorescence protein with NLS signal for the visual control of the promoter activity.

Packaging Lenti-X Vectors into lentiviral particles and virus tittering: The lentiviral construct, pLVX-TetOne-hPARG-mClover3-NLS-Puro was packaged into lentiviral particles using Lenti-X Packaging Single Shots Protocol from Clontech/Takara. Lentiviral vector stocks were collected and concentrated using the Lenti-X Concentrator Protocol (Clontech/Takara). Lenti-X virus stocks were tittered using puromycin selection with HT1080 cells; the titer of virus corresponded to the number of colonies generated by the highest dilution multiplied by the dilution factor.

ccRCC cell lines were infected with lentivirus, selected with puromycin and single clones were generated using FACS sorting or with cloning rings. Clones with the highest fold change of hPARG induction vs. control detected with Western blotting were selected for propagation and further experiments.

### 2.11. Western Blotting

Proteins from total cell lysates were separated by SDS-PAGE and transferred to a PVDF membrane. All blots were blocked in 5% dry milk, 0.1% Tween 20 (Millipore-Sigma P2287), then were probed by incubation with the following primary antibodies at 4C: rabbit monoclonal anti-hPARG (1:2000, Cell Signaling Technology, Danvers, MA, USA, D4E6X), mouse anti-hPARG (1:2000, Abcam, Cambridge, UK, ab169639), mouse monoclonal anti-pADPr (1:150, Santa Cruz Biotechnology, Santa Cruz, CA, USA, 10H), rabbit anti-PARP-1 (1:1000, Abcam, ab32138), mouse monoclonal anti-βtubulin (1:20000, Millipore-Sigma, B512), anti-Ecadherin (1:2000, Cell Signaling Technology, #3195), anti-β -Actin (1:10000, Mouse monoclonal, Millipore-Sigma, #A5441), anti-PARP-1 (1:1000, Cell Signaling Technology, #9532), or rabbit anti-pADPr binding reagent (1:2000, MABE1031, Millipore-Sigma). Membranes were then washed three times with 1X PBS with 0.1% Tween 20 and subsequently incubated with corresponding secondary antibodies (1:5000, PerkinElmer, Waltham, MA, USA) for 45 min at RT. Immunoblot exposure was done by adding HyGlo chemiluminescent HRP antibody detection reagent (Fisher Scientific, Waltham, MA, USA, NC9515009), and the imaging and densitometric analysis were conducted using a Licor Image machine and Image Studio v5.2.

### 2.12. Cell Cycle Analysis

ccRCC cells with TET-On PARG lentiX construct were plated in 6-well culture dishes in triplicates and doxycycline induction (500 ng/mL) was maintained for 72 h. Then cells were detached from plastic, fixed with ice cold 70% ethanol for 30 min and stained with FxCycle™ PI/RNase Staining Solution (Thermo Fisher, Waltham, MA, USA). Flow analysis was performed on FACSymphony A3 (BD Bioscience, Franklin Lakes, NJ, USA) and cells on G0/G1, S, and G2/M stages were calculated using FlowJo v10.8 software.

### 2.13. Apoptosis Level Evaluation in ccRCC Cells

ccRCC cells with TET-On PARG lentiX construct were plated in 6-well culture dishes in triplicates for each treatment group: DMSO only for 72 h, 500 ng/mL doxycycline induction for 72 h, 7.5 μM rucaparib for 48 h. Cells were detached from plastic with Accutase solution (StemCell Technologies, Vancouver, BC, Canada), fixed with freshly made 4% PAF for 15 min at room temperature, blocked and permeabilized with 0.7% Tween-20, 5% normal goat serum (Thermo Fisher), and 1% bovine serum albumin (Millipore-Sigma) on PBS for 45 min at room temperature. Cells were stained with primary rabbit antibodies against cleaved Caspase3 (1:300, Asp175, #9661S, Cell Signaling) diluted in blocking solution at 4 °C overnight and secondary anti-rabbit Alexa568 (1:800, Invitrogen) for 30 min at room temperature. Negative control with only secondary antibodies and positive control with cells treated for 8 h with 150 μM H_2_O_2_ were used. Flow analysis was performed on FACSymphony A3 (BD Bioscience) and analyzed using FlowJo software.

### 2.14. Clonogenic Assay

ccRCC/pLVX-TetOne-hPARG-Puro cells were seeded in 6-well plates in 500 cells/well concentration in triplicates with or without 500 ng/mL doxycycline. The cells were grown until colonies of around 50 cells were detectable. Cells were stained with DNA staining solutions containing OliGreen (1:4000, Invitrogen, Waltham, MA, USA) or propidium iodide (1:1000, Thermo Fisher) and counted.

### 2.15. Colony Formation Assay in Soft Agar

Colony-formation assay was performed according to published protocol with modifications [18]. Bottom layer with 0.6% agar in culture media was solidified in wells of 6 well plates. Cells were moved to single cell suspension and diluted in 0.3% agar media with or without of 500 ng/mL doxycycline in concentration 2000 cells per well in triplicates. Feeder layer with fresh media in 0.3% agar suspension was added every three days. Colonies images were captured with Cytation3 machine and counted.

### 2.16. Immunofluorescence Staining

ccRCC/pLVX-TetOne-hPARG-Puro cells were grown on 4 well Lab-Tek™ Chamber Slides (Thermo Fisher) with or without 500 ng/mL doxycycline for 72 h, washed with PBS, fixed with 4% paraformaldehyde for 10 min, and permeabilized with 0.3% TritonX100 for 20 min at room temperature. Fixed cells were blocked in 5% normal goat serum (Thermo Fisher) on 0.1% TritonX100 and PBS for 1 h at room temperature and then stained with primary rabbit monoclonal anti-PARG antibodies (1:500, D4E6X, Cell Signaling) or anti rabbit anti-pADPr binding reagent (1:200, MABE1031, Millipore-Sigma) on blocking solution at 4 °C overnight. Secondary anti-rabbit Alexa568 antibodies (1:800, Invitrogen) diluted on blocking solution were used for final 1 h staining at room temperature. Slide wells were removed and cells were mounted into Vectashield mounting media (Vector Laboratories, San Francisco, CA, USA).

### 2.17. RNA-Seq Analysis

PNX0010/pLVX-TetOne-hPARG-Puro cells were grown with or without 500 ng/mL doxycycline for 72 h and RNAs were extracted RNeasy kit (Qiagen, Hilden, Germany). The quality of RNA was determined by Bioanalyzer, the RNA integrity number for all RNA samples ranged from 9.9–10. RNA library preparation and 150 bp paired-end sequencing were performed at Novogene Sequencing laboratory. mRNA was enriched via Poly(A) selection. Novogene libraries were prepared using NEB’s Ultra II RNA library kit and sequencing was performed on the NovaSeq 6000 system (Illumina, San Diego, CA, USA). Illumina FQ sequencing files were imported into CLC Genomics Workbench version 12.0. The reads were trimmed to remove adapters then the trimmed reads were mapped to the human genome using CLC’s default parameters (mismatch cost = 2, insertion cost = 3, deletion cost = 3, length fraction = 0.8, similarity fraction = 0.8, auto-detect paired distances, maximum number of hits per read = 10). All samples have a minimum of 50 million reads with 92% or greater mapping in pairs to the genome. At least 97% of the mapped reads mapped to genes; *n* = 3 for both treatment groups. Differential expression for RNA-seq analyses was performed in CLC Genomics Workbench 12.0 on the gene-level expression tracks using whole transcriptome RNA-seq with TMM normalization. Differential expression due to treatment (PARG+ or control) was tested with the comparison against the control group. 

Genes with maximum group expression value >1 and FDR corrected *p*-value < 0.05 were subjected to STRING analysis (STRING string-db.org (accessed on 01 May 2021)) for multiple proteins, proteins with fold change as ranks, and for ingenuity pathway analysis (IPA). Gene ontology analysis was run on up- and down-regulated genes (GeneOntology geneontology.org (accessed on 30 July 2021)) powered by PANTHER.

### 2.18. Quantitative PCR (qPCR)

Total RNA was extracted from cells treated as indicated with RNeasy kit (Qiagen), contaminating genomic DNA was removed by the g-column. cDNA was obtained by reverse transcription using M-NLV reverse transcriptase (Invitrogen). Relative quantitative PCR was performed using SYBR Green Master Mix (Bio-Rad, Hercules, CA, USA) and ABI StepOne Plus real-time PCR system (Apply Biosystems, New York, NY, USA). Primers used are listed in Appendix A. For all qPCR experiments, gene expression levels were normalized to the human housekeeping gene GAPDH and averaged from triplicates. The relative expression levels of the genes were measured using 2^−ΔΔCt^ method. *n* = 3 biological replicates for each treatment group.

### 2.19. Statistical Analysis

Statistical analysis was performed using a two-sided Student’s *t*-test. A *p*-value of <0.05 was considered statistically significant.

## 3. Results

### 3.1. ccRCC Malignancy Is Associated with Aberrations in the Regulation of pADPr Turnover

PARPs catalyze the formation of pADPr; therefore, PARP activity can be assessed by measuring the level of pADPr. Recent studies reveal that pADPr levels are significantly up-regulated in a variety of non-urological cancers and that pADPr accumulation is associated with poor prognosis for cancer patients [19,20]. To examine the status of PARP signaling in ccRCC tumors, we analyzed the expression levels of pADPr in a panel of ccRCC cell lines. When compared to normal kidney epithelial cells, all tested ccRCC cell lines demonstrated the misregulation of pADPr pathway components. The most severe misregulation of pADPr turnover was found in ccRCC cell lines expressing augmented levels of PARP-1 protein (Figure 1A). Although pADPr levels do not entirely correlate with the levels of PARP-1 protein expression in tumor cell lines, all examined tumor cell lines demonstrated significantly increased pADPr expression compared to normal kidney epithelial cells (NK677) (Figure 1A, Appendix A). To gain further insights, we examined the expression levels of pADPr in clinical specimens of ccRCC and corresponding normal kidney tissue. As demonstrated in Figure 1B and Appendix A, pADPr expression was significantly augmented in two out of four ccRCC specimens compared with the corresponding normal kidney tissue samples. Notably, the most intense expression of pADPr was detected in ccRCC tissue specimens obtained from metastatic lesions. The results of these experiments were validated by immunostaining of primary and metastatic tumor specimens, as well as normal kidney tissue sections, using anti-pADPr antibody (Figure 1C). Moreover, when we utilize the TCGA PanCancer Atlas for relative mRNA expression in different tumors to corresponding normal tissue, the mean expression of PARG mRNA in ccRCC was the lowest and most severely down-regulated across all tested cancer types (Appendix A).

This extreme accumulation of pADPr and its turnover deficiency in ccRCC cell lines and primary tumors encouraged us to test whether the reduction of pADPr level would have an effect on ccRCC malignancy. We selected two approaches to decrease pADPr: inhibit its synthesis by PARP-1 inhibitors and activate its hydrolysis by PARG overexpression.

### 3.2. pADPr Reduction by PARP-1 Inhibitors Suppresses the Growth of ccRCC Tumors In Vitro and In Vivo

We selected ccRCC cell lines with the highest pADPr levels to study their response to PARP-1 inhibition (Figure 1A): four established ccRCC cell lines, 786-O, 769-P, SK-RC-45, and SK-RC-26b, and short-term cultured primary ccRCC cell line PNX0010 that were derived and established in our laboratory and have known low number of passages [21,22]. First, we assayed the proliferation rates of the studied ccRCC cell lines and NK677 normal kidney epithelial cells under PARP-1 inhibition with the NAD-competitive PARP-1 inhibitors olaparib (ZD-2281, Lynparza AstraZeneca, Cambridge, UK) and rucaparib (AG014699, Rubraca, Clovis Oncology, Boulder, CO, USA) (Figure 2D, Appendix A).

All cell lines demonstrated diminished proliferation rate (Figure 2A,B,E). However, both olaparib and rucaparib also suppressed the viability of normal kidney cells (Figure 2A,B,E).

These promiscuous effects on both normal and cancer cells could have resulted from the competitive nature of NAD-like PARP-1 inhibitors olaparib and rucaparib. Inhibition of PARP-1 by competing with NAD tends to affect a number of different metabolic processes. To overcome the limitation of NAD-like PARP-1 inhibitors, we developed a novel class of non-NAD-like PARP-1 inhibitors based on the molecular mechanism underlying PARP-1-dependent transcription with non-NAD like structure [23,24,25]. 5F02 has no obvious structural homologues among components of eukaryotic enzymatic pathways and believed to disrupt PARP-1 binding and activation by histones [26]. Non-NAD-like nature minimizes off-target effects and ensures greater specificity. To evaluate the therapeutic potential of these inhibitors, we compared the effect of non-NAD-like PARP-1 inhibitor 5F02 and NAD-like olaparib on the viability of patient-derived PNX0010 ccRCC cells and NK677 normal kidney epithelial cells. As demonstrated in Figure 2A, 5F02 suppressed the growth of PNX0010 cells with higher specificity and efficacy when compared to olaparib. 5F02 eliminated PNX0010 cells (IC50—1.8 μM) with almost no cytotoxicity to normal cells, while olaparib suppressed viability of both normal (IC50—14.7 μM) and cancer cells (IC50—7.2 μM). 5F02 also demonstrated its superior antitumor activity in the clonogenic assay compared to that of olaparib (Figure 2B).

In light of these encouraging in vitro data, we examined the antitumor activity of non-NAD-like inhibitors using ccRCC xenograft tumors established from PNX0010 ccRCC cells. As demonstrated in Figure 2C, animals treated with the non-NAD-like inhibitor 5F02 showed a significant inhibition of tumor growth relative to control animals, i.e., animals treated with the NAD-competitive PARP-1 inhibitor olaparib and animals treated with the multitargeted tyrosine kinase inhibitor sunitinib, a standard agent for the treatment of metastatic RCC [27] (Figure 2C). Moreover, our studies demonstrate that 5F02 inhibits PARP-1 activity in xenograft RCC tumors with higher efficacy when compared to olaparib (Figure 2D, Appendix A). Importantly, treatment with 5F02 was well tolerated by all animals with no apparent signs of toxicity. Staining tumor section with mitotic cells marker phH3Ser10 demonstrated reduced proliferation rate for 5F02 but not for olaparib and sunitinib-treated xenografts (Appendix A). We detected no statistical difference of apoptotic rate for 5F02 experimental and control groups (Appendix A).

### 3.3. pADPr Reduction by PARG Overexpression Suppresses the Growth of ccRCC Tumors In Vitro

Even highly specific inhibitors could cause non-selective off-target effects. To check whether the pADPr pathway disturbance solely suppresses malignancy, we based our experiment on the fact that PARG is a solo enzyme to effectively hydrolyze and break down pADPr chains [10]. We created PARG TET-On ccRCC cell lines that can overexpress PARG under doxycycline induction (Figure 3A). 786-O, 769-P, SK-RC-45, SK-RC-26b, and PNX0010 cell lines were permanently transduced with a lentivirus construct, and single clones, or a mixed population, were generated. Upon doxycycline induction, all created cell lines exhibited PARG overexpression and reduction of pADPr level (Figure 3B, Appendix A). Importantly, PARG overexpression does not affect viability and proliferation rate of normal kidney cells NK677 (Appendix A).

Cell cycle assay demonstrated decreased mitotic rate under PARG overexpression with more cells at G0/G1 stage for all studied ccRCC cell lines (Figure 3D, Appendix A). We tested the ability of PARG overexpressing ccRCC cell to form colonies on plastic. This clonogenic assay showed the reduction of number of colonies for all tested cell lines (Figure 3C, Appendix A). Moreover, the colonies for no doxycycline induced cells generally contained more cells (unpublished observation). Then we decided to test how PARG overexpression will affect the true malignant cells ability to form colonies without extracellular matrix and grown cells in soft agar. Surprisingly, when proliferation was not extremely affected and clonogenic assay on plastic demonstrated 50–70% reduction rate, PARG overexpression much severely reduced the number of colonies in soft agar in unanchoring condition for PNX0010, 789-O, SK-RC-45, and SK-RC-26b (Figure 3E,F, Appendix A). 769-P cells were unable to generate any colonies for both control and experimental condition. Thus, it indicates that ccRCC cells lost tumorigenicity under PARG overexpression and were unable to proliferate at surface-free state, the unique trait of cancer cells.

PARP-1 is considered to play important role in DNA damage reparation. The increase in cell death due to unpaired reparation could be the explanation for the observed effects under PARP-1 activity modulation. To address this possibility, we measured the apoptosis level in studied ccRCC cell lines for PARP-1 inhibition and PARG overexpression conditions. Apoptosis level was low in all experimental groups, close to control group and did not demonstrate any significant changes (Appendix A).

### 3.4. Lipid/Cholesterol Biosynthesis and Interferon Signaling Pathways Are Up-Regulatedand Angiogenesis and Tumor Growth-Related Genes Are Down-Regulated under PARG Overexpression in PNX0010 ccRCC Cells

To identify the transcriptional changes that could underlie the observed phenotypes, we performed RNA-seq analysis of doxycycline-treated transduced PNX0010 cells overexpressing PARG and compared the results with those of untreated transduced cells. Control and experimental groups were done in triplicates and differential expression analysis was performed on gene-level expression tracks after TMM normalization with fold change calculation statistical test based on a negative binomial generalized linear model. FDR-corrected *p*-value was calculated and only genes with value less than 0.05 were considered as significant and were used for the analysis. We excluded extremely low expressing genes with maximum group expression value less than 1. 99 and 118 genes were significantly down- and up-regulated, respectively, (Appendix A), among them, 28 and 19 were differentially down- and up-regulated with fold change difference greater than 1.5 (Figure 4, Appendix A).

We subjected all differentially expressed genes with FDR-corrected *p*-values less than 0.05 and expression value greater than 1 for STRING analysis. To detect most affected pathways, we filtered enriched GO terms as following: gene count in pathway more than 7, enriched score more than 0.7, and FDR *p*-value less than 0.001. Total of 30 enriched pathways were identified (Appendix A) related to three categories: regulation of lipid metabolic process, interferon signaling, and angiogenesis. These three pathways were clustered together in STRING dot graph (Figure 4A). Panther analysis for biological pathways up- and down-regulated genes separately showed that lipid metabolism and interferon signaling pathways became up-regulated with enrichments scores more than 6.04 and 4.34 correspondingly and angiogenesis became down-regulated with enrichment score 7.2 (Figure 4, Appendix A).

IPA analysis revealed that 35 out of 51 enzymes responsible for the conversion of acetyl-CoA to cholesterol became misregulated as the whole pathway became up-regulated (Figure 4, Appendix A). Two main regulators of cholesterol biosynthesis are the transcriptional factors sterol regulatory element-binding proteins 1/2 (SREBP1/2), which activate this pathway, and their suppressor insulin-induced gene 1 (INSIG1), which correspondingly inhibits the pathway [28,29,30]. When this pathway is up-regulated, such as in PNX0010 cells under PARG overexpression, the increased SREBP1/2 level or decreased INSIG1 level is expected. Surprisingly, SREBP1/2 level was not affected and INSIG1 expression was up-regulated by 1.8-fold. These findings indicate that either mRNA level does not correspond to protein level for this regulator or that pADPr regulates transcription of cholesterol biosynthesis pathway genes in some other way. Not only cholesterol synthesis, but its uptake was also enhanced: low-density cholesterol receptors (LDLR) became up-regulated by 1.5-fold. Fatty acid synthesis is also increased by up-regulation of the rate-limiting enzymes fatty acid synthase (FASN) (1.5-fold change) and stearoyl CoA desaturase 1 (SCD1) (1.8-fold change). As important components of cell membranes, along with cholesterol, fatty acids play a vital role in cancer development.

Another up-regulated pathway under PARG overexpression in PNX0010 cells is interferon response (Figure 4A, Appendix A). For interferon α response, 17 genes are differentially expressed with 16 up-regulated. 

The third cluster of differently expressed genes generated by STRING analysis was related to angiogenesis GO terms (Figure 4A) and enriched genes were among the down-regulated genes (Appendix A). 

To identify what genes that became misregulated in PNX0010 cells overexpressing PARG are known to be related to cancer growth, we applied our differently expressed genes to IPA software 2020 Fall release for “cancer” disease phenotype. In total, IPA analysis of transcriptome predicted down-regulation of “growth of tumor,” “growth of solid tumor,” and “growth of malignant tumor” phenotypes with enrichment score 2.4 (Appendix A).

Among the top differentially expressed genes we discovered three inhibitor of DNA binding 1-3 (ID1-3) genes (Figure 4B). These genes are known to be key developmental regulators intensively associated with tumor progression and ID1 is a negative prognostic marker for ccRCC [31]. Other cancer progression-related genes that became differently expressed in PNX0010 cells under PARG overexpression are serpin family E member 1 (SERPINE1), colony-stimulating factor 2 (CSF2 or GM-CSF), ubiquitin specific peptidase 10 (USP10), and growth differentiation factor 15 (GDF15). 

### 3.5. PARG Overexpression and PARPs Inhibition Similarly Affect Gene Expression

Because both PARG overexpression and PARP-1 inhibition lead to pADPr level reduction, we anticipate seeing the involvement of common pathways and similar gene misregulation in these two approaches. First, we selected the down-regulated genes at PNX0010 under PARG overexpression (Figure 5A) that are related to cancer development: *ID1*, *ID2*, *ID3*, and *SERPINE1* and performed qPCR analysis for 786-O, 769-P, SK-RC-45, and SK-RC-26b cell lines treated with the 10 uM PARP-1 inhibitor rucaparib for 24 h (Appendix A). For all studied cell lines, we have shown that the reduction in pADPr level leads to down-regulation of ID genes with 786-O cell line response less intense than that of the other cell lines (Figure 5B). *SERPINE1* was also down-regulated but reached statistical significance only for 786-O and SK-RC-45 cell lines (Figure 5B).

Upregulation of lipid biosynthesis genes in PNX0010 cells overexpressing PARG encouraged us to test whether we would see similar changes upon PARP-1 inhibition. Expression of *SCD1* (fatty lipid metabolism) and *INSIG1* (suppressor of cholesterol biosynthesis) genes was evaluated. Interestingly, we found just the opposite effect of PARP-1 inhibitors on the expression of these genes. Both genes were significantly down-regulated under rucaparib treatment in 786-O, 769-P, SK-RC-45, and SK-RC-26b cell lines (Figure 5B).

## 4. Discussion

Aberrations in the VHL gene are the most important risk factors for the development of RCC, especially ccRCC [2,3]. In sporadic ccRCC, about 70% of all tumors harbor biallelic inactivation of VHL through mutation, deletion, or hypermethylation of promoter that leads to the loss of its expression [2,8]. ccRCC is a highly lethal disease with incidence on the rise [32]. While significant advances in both extirpative and systemic approaches have been achieved, patients with metastatic or reoccurring cancer are still at high risk of death from ccRCC. Therefore, the development of new therapeutic strategies for advanced kidney cancer represents a significant challenge. The clinical potential of PARP-1 inhibitors has been increasingly recognized over the past two decades, prompting intensive research on their therapeutic application [20,33,34]. Several clinical trials are ongoing to study the effect of PARP-1 inhibitors on participants with renal cell carcinomas. However, a more targeted approach to treat ccRCC is urgently needed.

The high level of endogenous pADPr appears to be the best predictor of tumor responsiveness to PARP-1 inhibition. Importantly, pADPr is overaccumulated in a variety of cancers, and its build-up has been associated with poor prognosis for cancer patients [19,20]. We have examined a large number of cancer and normal cells of various origins (kidney, breast, ovarian, prostate, and leukemia) for aberrations in the poly(ADP-ribosyl)ating pathway, including PARP-1 protein and PARG enzyme [12,22,25,35]. When compared to normal cells, all tested cancer cell lines demonstrated the misregulation of pADPr pathway components. The most severe changes of pADPr turnover were found in ccRCC cell lines expressing augmented levels of PARP-1 protein compared with normal kidney epithelial cells (Figure 1A) and in metastatic ccRCC clinical specimens (Figure 1B). Taken together, these findings suggest that high level of endogenous pADPr may serve as a potential biomarker for sensitivity to PARP-1 inhibitors. pADPr turnover is regulated by the enzyme PARG, which degrades pADPr to free ADP-ribose and AMP. Notably, PARG expression was significantly reduced in ccRCC cell lines compared to normal kidney epithelial cells (Figure 1A). Moreover, PARG expression was substantially reduced in all tested metastatic ccRCC samples (Figure 1B). Thus, low levels of PARG enzyme may serve as another potential biomarker for sensitivity to PARP-1 inhibitors.

We investigated how disrupting pADPr turnover and reducing its level would affect the tumorigenic ability of different established and primary ccRCC cell lines. To identify direct effects and escape the risk of “off-target” effects of chemical compounds, we applied a three-way approach. Specifically, we utilized classical NAD-like PARPs inhibitors, a new class of non-NAD-like inhibitors and inducible overexpression of the unique pADPr degrading enzyme PARG. All three approaches demonstrated excellent results and reduced the viability, proliferation rate, and colony-formation ability of ccRCC cell lines (Figure 2 and Figure 3). However, while the non-NAD-like inhibitor 5F02 had no effect on the control kidney cell line, classical NAD-like inhibitors did have a deleterious impact on normal cell viability (Figure 2B). In our experimental setup, PARP-1 inhibitors reduced pADPr level more efficiently than the PARG overexpression method (Appendix A). That could explain the stronger effect in a set of tumorigenicity tests. However, mild PARG overexpression at physiological level is clinically more relevant and corresponds to what could be achieved in patients. From this aspect, while PARG overexpression did not cause active ccRCC cell death or reduce the proliferation rate less extensively than PARP-1 inhibitors, it is even more encouraging that the colony formation ability in soft agar at unanchored state was extremely disrupted (Figure 3D,F, Appendix A). Notably, this is the main cancer cell trait that distinguishes cancer cells from normal cells, i.e., the ability to divide without the presence of extracellular matrix.

Transcriptome analysis of the PARG-overexpressing primary PNX0010 ccRCC cell line uncovered transcriptional changes affected by pADPr reduction and could explain the observed phenotypes. First, several genes known to be connected to the cancer state became down-regulated (Figure 4B). Among them is the class of ID genes, in particular ID1-3, since these genes were significantly down-regulated under both PARG overexpression and PARP-1 inhibition. IDs are members of the helix-loop-helix (HLH) protein family and act as direct or indirect negative regulators of basic HLH transcription factors involved in many developmental events and stem cell pluripotency keeping (reviewed in [36]). The essential role of these proteins for proliferation, tumor progression, angiogenesis, and invasion was shown for more than 20 types of cancer [31,36,37,38]. Critically, ID2 was proven to be a direct target for HIF1a transcriptional factor [39]. Recent studies suggest that ID1 is a strong prognostic biomarker for ccRCC and link its up-regulation with poor survival and high probability of tumor metastasis [31].

Genes involved in angiogenesis pathway were down-regulated upon PARG overexpression that could decrease blood supply induction and inhibit tumor growth. Even though cancer cells possess extraordinary ability to divide and survive, the growth of a tumor ultimately requires a blood supply. Tumor cells express stimulatory factors to initiate vascular growth by attracting and activating cells from within the microenvironment of the tumor. ccRCC is one of the most highly vascular tumor type, which is driven by intensive release of vascularization factors in response to VHL mutation and hypoxia pathway activation [40]. Targeting angiogenesis using multitargeted tyrosine kinase inhibitors (TKIs), has resulted in doubling of progression-free survival and significant gains in overall survival, thereby notably changing the treatment paradigm of advanced kidney cancer.

Second, lipid/cholesterol synthesis became significantly up-regulated in PARG-overexpressing PNX0010 cells (Figure 4C). This is noteworthy because the development of ccRCC is more reliant on metabolic changes than other tumors [41]. The term “clear-cell RCC” itself originates from the clear (empty) microscopic appearance of the cytoplasm after the lipids are removed in the process of fixation. This happens because tumor-driven VHL mutation leads to the activation of the HIF1/2α pathway and downstream pathway permutation [42,43,44]. Many of these pathways are metabolic, in particular, pathways related to glycolysis, fatty acid, and cholesterol synthesis [45,46]. ccRCC tumors rely on these extensive metabolic changes more than other tumors and utilize them to support their malignant growth. It was shown that ccRCC has a more pronounced Warburg effect than other tumors, meaning enhanced glycolysis and suppressed glucose oxidation [41].

Any disruption of ccRCC metabolism can also disrupt its continuous growth. Different approaches to disrupt ccRCC metabolism were successfully applied to inhibit malignancy and uncontrolled cell growth [41,47]. Interestingly, the effect of PARG overexpression and PARPs inhibitors was different on the transcription of some lipid synthesis-controlling genes (Figure 5A,B). These controversial effects on different gene expression could be explained by the different level of pADPr suppression by rucaparib and PARG overexpression or the NAD-like nature of rucaparib. NAD is an abundant and ubiquitous molecule used by numerous enzymes, and the main off-target effects from PARPs inhibition by NAD competitors are supposed to primarily affect metabolic processes [48]. 

Finally, the interferon response pathway was up-regulated in PNX0010 cells overexpressing PARG. A interferons are pleiotropic cytokines, extensively used in the treatment of patients with certain types of cancer, affecting tumor growth by different mechanisms. Recombinant IFNα2 became the first human immunotherapeutic agent approved by the FDA for RCC cancer treatment. However, low efficacy rate (5–20%) and unpleasant flu-like side effects are the major limiting factors for its clinical application [49,50]. In our experiments, the possible up-regulated response to interferons could be beneficial owing to the increased internal sensitivity to endogenous interferons, revealing new insights allowing to work toward combinational PARP-1 inhibitor/interferon therapy with lower interferon dosage.

Other cancer progression-related genes expressed differently under PARG overexpression in PNX0010 cells. Downregulated gene SERPINE1 encodes plasminogen activator inhibitor-1 (PAI-1) protein, which plays an important role in the regulation of extracellular matrix remodeling. Initial research mainly focused on its role in thrombosis, but current studies suggest that PAI-1 also plays a critical role in tumorigenesis of various types of cancers. Emerging investigations of PAI-1 favored its pivotal implications for cell migration, invasion, and tumor vascularization, elucidating the tumor-promoting roles [51,52].

Down-regulated GM-CSF is a cytokine that promotes stem cells to become granulocytes and monocytes, but also affects more broad ranges of cells, stimulates immune response and inflammation. The GM-CSF involvement in cancers is still controversial. While treatment with GM-CSF is mainly used for cancer patients to overcome chemotherapeutic neutropenia and marrow damage, plenty of studies successfully use it in anti-cancer vaccines to increase the anti-tumor immune response [53]. It is an important cytokine in the activation of dendritic cell formation and the enhancement of dendritic cell activity [54]. Controversially, other studies suggest that GM-CSF can impair antitumor immune responses and has an immunosuppressive effect in the blood and tumor microenvironment [55,56,57]. Prognostic value of GM-CSF expressional changes reveal cancer-type-dependent impact and its high level indicated poor prognosis in RCC (*n* = 533, HR:2.836, 95%CI: 2.019–3.983, *p* = 0.001) [58]. Moreover, in some cases, ccRCC is shown to produce high amount of GM-CSF itself and this trait could be used as a biomarker for tumor reoccurrence [59,60].

USP10, the most abundant deubiquitinase in PNX0010 cells, was shown to be highly specific for p53 in ccRCC cells [61], acting as a tumor suppressor and becoming up-regulated under PARG overexpression. GDF15 is another gene that became down-regulated. It is a member of the TGF-β superfamily, and GDF15 expression was demonstrated for various human cancers [62,63].

## 5. Conclusions

Here we utilized new approaches to reduce the pADPr level in ccRCC cells and demonstrated the prominent antitumor effect of these “back-to-normal” interventions. Both, overexpression of PARP-1 antagonist, PARG, or the treatment with PARP-1 inhibitors reduced viability and clonogenic potential of ccRCC cell lines and suppressed growth of ccRCC xenograft tumors. Transcriptome analysis linked observed phenotype with changes in gene expression levels for lipid metabolism, interferon signaling, and angiogenesis pathways along with the changes in expression of key cancer-related genes.

PARP-1 inhibitors are successfully used in clinic for treatment of different types of cancers. Numerus evidence suggest the defects in DNA damage reparation as a main mechanism of action. Our work supports the idea that reduction of pADPr level and PARP-1 activity is also critical for anti-tumor effects as the PARG overexpression approach demonstrated similar phenotype.

## Figures and Tables

**Figure 1 cancers-13-04973-f001:**
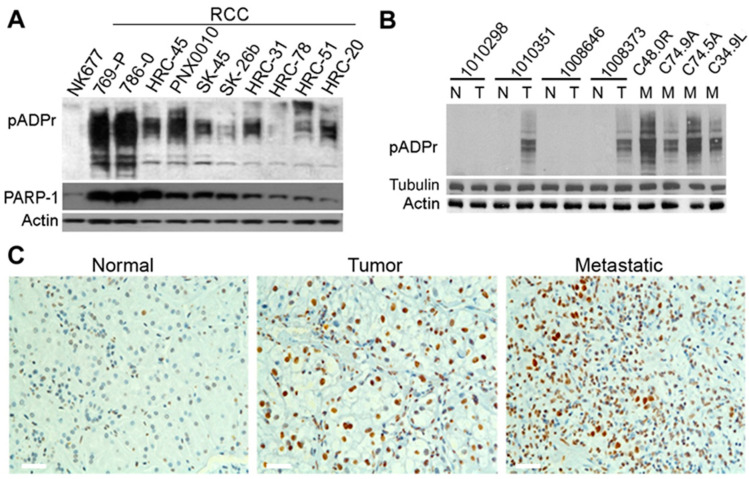
ccRCC cell lines and tumors affected by pADPr turnover. Western blot analysis for pADPr and PARP-1 in ccRCC cell lines (**A**) and patient-derived tumors (**B**). “N,” “T,” and “M” indicate “normal,” “tumor,” and “metastatic” samples, respectively. They were obtained from the same patients and are presented in corresponding order. All studied ccRCC samples demonstrate a high level of pADPr. Actin and Tubulin levels are shown as a loading control. (**C**) Immunohistochemical staining of cryosectioned normal kidney, ccRCC tumor, and metastatic samples from patients reveals significant accumulation of pADPr (brown) in nucleus of tumor and metastatic cells. Representative images are shown. White scale 20 µm. The uncropped Western Blot images can be found in Appendix A.

**Figure 2 cancers-13-04973-f002:**
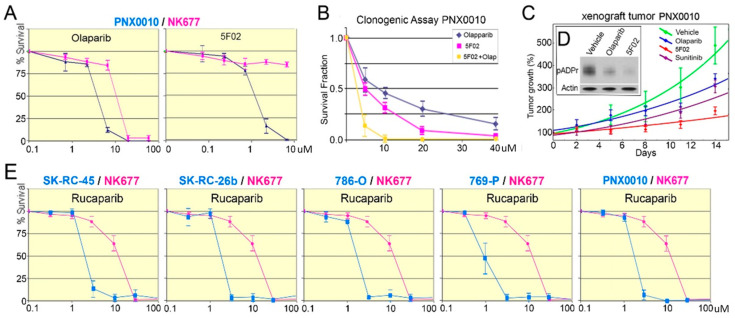
PARP-1 inhibitors affect ccRCC viability, proliferation and xenograft growth. (**A**) NAD-like and non-NAD-like PARP-1 inhibitors olaparib and 5F02, respectively, affect patient-derived ccRCC PNX0010 cell line viability. 5F02 eliminated PNX0010 cells (IC50—1.8 μM) with almost no cytotoxicity to normal cells, while olaparib suppressed viability of both normal (IC50—14.7 μM) and cancer cells (IC50—7.2 μM). (**B**) PARP-1 inhibitors suppress malignancy potential of cancer-derived cells PNX0010. Calculation of cell survival rate was based on clonogenic assays. Cells were plated into 24-well plates. Cells were allowed to adhere overnight and were treated with a non-NAD-like inhibitor (5F02) (magenta), olaparib (blue), and both (red) for 14 days. Colonies were counted and plotted on the graph. Inhibition of pADPr by olaparib and 5F02 is shown in Appendix A. (**C**) Tumor growth of PNX0010 xenografts is diminished by PARP-1 inhibitor treatments with olaparib (blue) and 5F02 (red), as well as the classical anti-ccRCC drug sunitinib (purple). Growth of tumor without any treatment is shown in green. The most profound effect is demonstrated for 5F02 compared to control group (*p* < 0.01) and olaparib (*p* < 0.05) group. (**D**) Western blot analysis shows the reduction of pADPr level in tumors upon olaparib and 5F02 PARPs inhibition. Actin level is shown as loading control. (E) Treatment of ccRCC cell lines 786-O, 769-P, SK-RC-45, SK-RC-26b and normal NK677 with PARP-1 inhibitor rucaparib demonstrates the potential reduction of proliferation rate. Rucaparib eliminated normal cells (IC50—15.2), SK-RC-45 (IC50—1.38 μM), SK-RC-26b (IC50—1.31 μM), 786-O (IC50—1.29 μM), 769-P (IC50—0.98 μM), PNX0010 cells (IC50—1.4 μM). The uncropped Western Blot images can be found in Appendix A.

**Figure 3 cancers-13-04973-f003:**
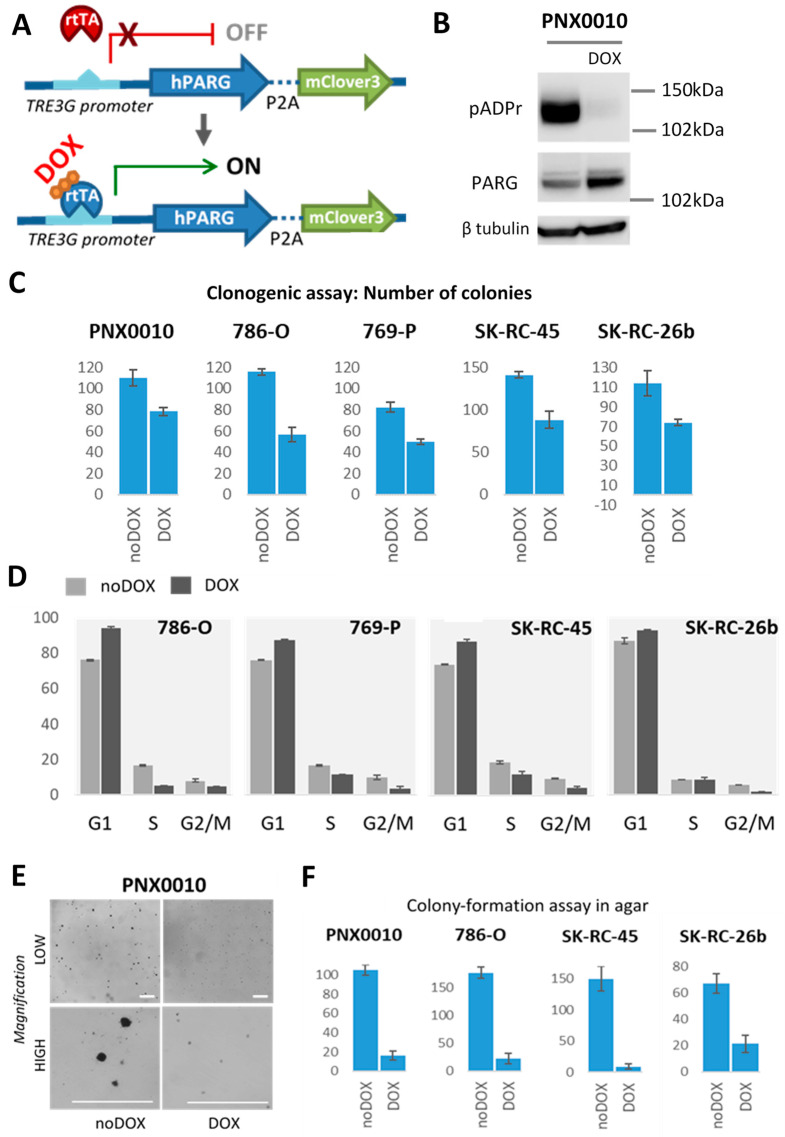
PARG overexpression reduces tumorigenicity of ccRCC cell lines. (**A**) Schematic representation of doxycycline inducible system for PARG overexpression experiment. HIS-tagged PARG cDNA was cloned into lentivirus plasmid pLVX-TetOne™-Puro (Takara) after doxycycline TET-On inducible promoter, followed by P2A cleavable mClover3 green fluorescence protein for expression visualization. Puromycin selection was utilized to expand positively transduced cells. (**B**) pADPr inhibition in PNX0010 cell lines overexpressing PARG under 500 ng/uL doxycycline stimulation for 72 h. Tubulin level is shown as a loading control. (**C**) cRCC cells were seeded at 500 cells per well at 6-well plate in triplicates and grown without/with 500 ng/mL doxycycline. Number of colonies were counted after 5–7 days of culture when the colonies reached the size around 50 cells/colony. ccRCC under PARG overexpression form less colonies when grown on plastic, *p*-value < 0.05. Sample wells with colonies for studied cell lines are shown in Appendix A. (**D**) ccRCC PARG Tet-On cell lines were grown for 72 h without/with 500 ng/mL doxycycline in triplicates and cell cycle analysis was performed with DNA propidium iodide staining and flow cytometry. Percentages of cells on each G0/G1, S, and G2/M stages were calculated with FlowJo software. Representative histograms are present in Appendix A. For all studied ccRCC cell lines PARG overexpression caused accumulation of cells at G0/G1 stage. (**F**) cRCC cells were seeded at 2000 cells per well at 6-well plate concentration in 0.3% agar in media without/with 500 ng/mL doxycycline. Fresh 0.3% agar was added each third day. Number of colonies were counted after 2 weeks of culture. 769-P cell line was unable to form colonies. Sample PNX0010 colonies at lower and higher magnification are shown in (**E**) and sample wells for studied cell lines are presented in Appendix A. White scale bar 150 um. ccRCC under PARG overexpression form less colonies when grown in an unanchored state in agar, (*p*-value < 0.05). The uncropped Western Blot images can be found in Appendix A.

**Figure 4 cancers-13-04973-f004:**
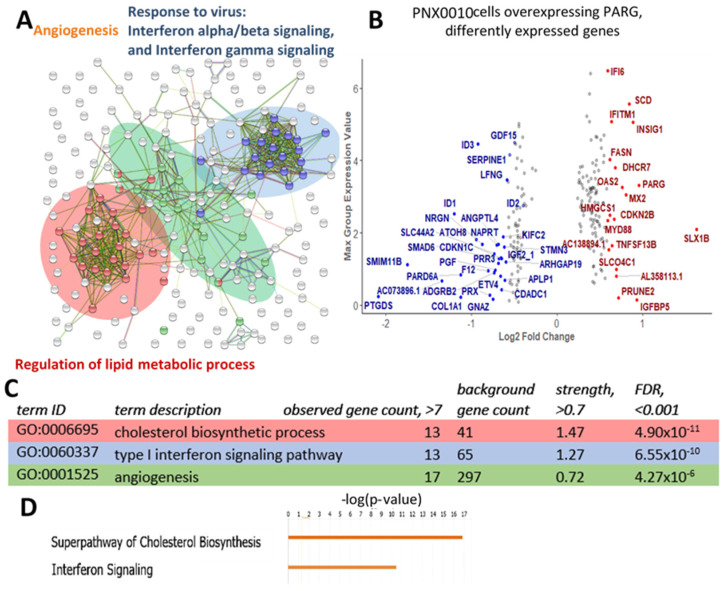
Lipid biosynthesis, interferon signaling, angiogenesis pathways, and several cancer-driving genes are affected under PARG overexpression in PNX0010 cell line. (**A**) Differently expressed genes with FDR-corrected *p*-value < 0.05 and maximum group expression value >1 were subjected to STRING analysis. Three main clusters were identified that are related to lipid metabolism, interferons signaling, and angiogenesis. (**B**) Differently expressed genes with FDR-corrected *p*-value < 0.05 and maximum expression value > 1 are plotted and genes with fold change >1.5 are named. (**C**) GO terms for selected overrepresented pathways (for the whole list see (for the whole list see Appendix A). (**D**) IPA analysis shows activation of cholesterol biosynthesis and interferon signaling pathways.

**Figure 5 cancers-13-04973-f005:**
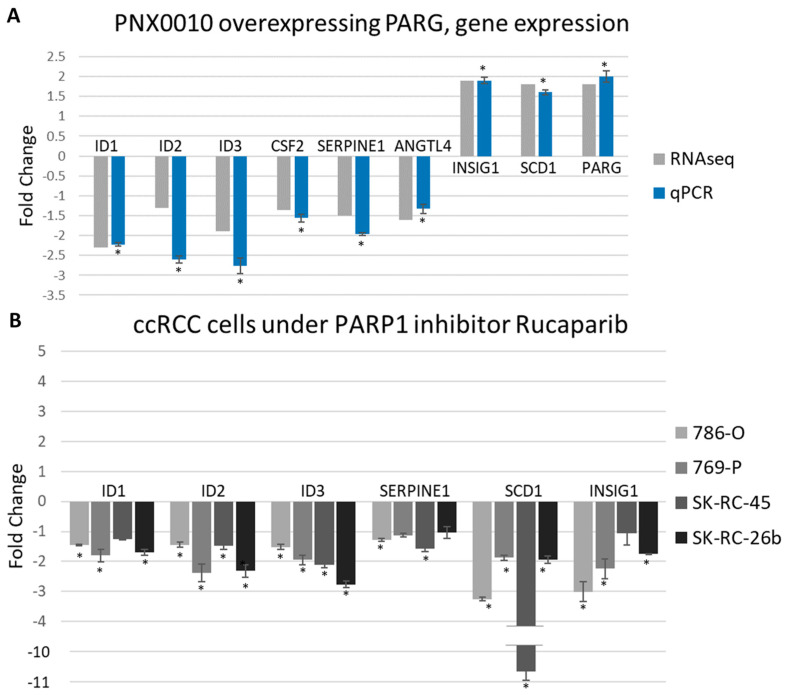
pADPr reduction by PARP-1 inhibition and PARG overexpression affects gene expression in similar way. (**A**) Changes in expression of selected genes measured by qPCR and RNAseq methods in PNX0010 cell line under PARG overexpression. Correspondence of two analyses is demonstrated. * *p*-value < 0.05 (**B**) qPCR analysis of genes from (A) in ccRCC cell lines under PARP-1 inhibition with 10 uM rucaparib for 24 h. Downregulation of cancer-related genes ID1-ID3 is shown for both PARG overexpression and PARP-1 inhibition. Opposite effect was shown for metabolism-related genes and colony-stimulation factor 2 coding gene. Serpine1 gene shows concordant results under PARP-1 inhibition and PARG overexpression for 786-O and SK-RC-45 cell lines. * *p*-value < 0.05.

## Data Availability

Transcriptome analysis is available at GEO, accession number GSE184354.

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
