# Peer review of "Poly(ADP)-Ribosylation Inhibition: A Promising Approach for Clear Cell Renal Cell Carcinoma Therapy"

_cancers, 2021, doi:10.3390/cancers13194973_

Round 1
Reviewer 1 Report
The authors in this work demonstrated that PARP-1 inhibitors could represent a potential therapeutic approach in ccRCC. Using preclinical models in vitro (cell lines and primary cultures) and in vivo (xenograft of ccRCC cell lines) they demonstrated that these inhibitors reduce tumor growth. However the paper is not acceptable in the following form because it needs important major revisions.
Major revisions
1) In results section (lanes 236-238) the authors postulated that “The most severe misregulation of pADPr turnover was found in ccRCC cell lines expressing augmented levels of PARP-1 protein (Figure 1A)” , but in figure 1 are not indicated the PARP-1 expression; only pADPr and actin protein levels are showed.
2) In results section (lane 243-250) the authors described the expression of pADPr in human tissue specimens. In materials and methods the authors should add a paragraph about the use of biological samples derived from patients, indicating that the study has been approved by an ethics committee and the patients have signed informed consent. Furthermore, the analysis was carried out on a restricted series (n = 4) and 2/4 express the protein compared to the normal counterpart. In my opinion they should increase the case series. There were also 4 tissues from metastases, but it is not clear whether they derive from the four patients analyzed or if they derive from other patients. Finally, in the material and methods section, the tissue immunostaining procedures should be added.
3) In section 3.2 (suppression of growth in vitro and in vivo by PARP-1 inhibitors), the authors, in the text write that they have treated some cell lines with Olaparib and Rucaparib (lane 270), but growth inhibition curves with 5F02 and Sunitinib appear in the figures. I suggest adding in the methods a section describing all drugs used in vitro and in vivo experiments, with the suppliers, the different preparations (diluent/stock concentration) and storage temperatures.
Regarding the in vitro experiments it would be appropriate to add a table with the IC50 values and the standard deviations of all inhibitors used on the in vitro models (normal renal epithelial cells, cell lines and primary culture). In addition, the inhibition curves in Figures 2A, B and E standard deviations are not represented and appear to be only representative experiments. How many experiments were performed? Each experiment was performed in triplicate? This information should be added in the methods section and legends. The growth curves of the 4 cell lines, of the culture primary and normal epithelial cells should be in a single graph, for each inhibitor (Olaparib, 5F02, Rucaparib). In fact it is unclear why Olaparib, 5F02 and their combination were used with PNX0010 cells, while for the other lines Rucaparib alone. Furthermore, also in Figure 2B there are no standard deviations.
The western blot is shown in supplementary figure S2 in which is demonstrated the downregulation of pADPr after treatment with Rucaparib. Authors should perform further western blot to verify protein downregulation in all cell lines treated with all inhibitors. Furthermore, supplementary figure 2 is not mentioned in the text.
Figure 2C. I would remove the western blot (panel D) and put it in supplementary figure X. Please, Indicate the abbreviation Med in the legend. Why was Sunitinib also used and not Rucaparib in vivo? Explain the choice of use of PNX0010 cells for the in vivo study. Are the others tumorigenic? Include representative images of the tumors (explanted or in mouse) in the supplementary date. The proliferative index (Ki67), apoptosis (i.e. tunnel assay) should be checked on explanted tumors. It is known that PARP-1 inhibitors induce apoptosis in several tumor models in vitro and in vivo.
Finally, in the figures and in the text, the SK-26b and SK-45 cell lines are sometimes referred to as SK-RC-26b and SK-RC-45. I suggest to standardize. Indicate in the methods, for in vivo studies, how many animals were used per cohort and how many experiments were carried out.
It would be advisable to verify whether PARP-1 inhibitors induce apoptosis in the treated-ccRCC cell lines. If the cells are blocked in the G1 phase, the inhibitors could be cytostatic and non-cytotoxic.
4) Paragraph 3.3: how many experiments were carried out for the study of the cell cycle for each cell line (figure 3C)? Are the plots with SD in Figure 3C the result of three experiments? Indicate this information in the legend and in the materials and methods section. I suggest removing figure 3B and in supplementary figure y show the representative histograms of the different phases of the cell cycle of all lines infected with the lentivirus carrying hPARG with and without doxycycline.
Figures 3D-E-F-G: also in this case indicate how many experiments have been carried out. Are the images D to F representative? Add in supplementary figure z the evaluation experiments of the clonogenic assay and colony-formation assay of all the lines and not just the PNX0010 cells.
Also in this cycle of experiments, the evaluation of apoptosis was omitted.
5) The validation of the modulated genes after PARG hyperexpression in PNX0010 should also be carried out on the RNA extracted from the other cell lines infected with the lentivirus carrying hPARG
Minor revisions
1) In the methods section you should indicate at what tumor volume the treatment of mice with inhibitors was started
2) In lane 220 the Table 2 of supplementary data is Supplementary Table S4.
Reviewer 2 Report
In the manuscript “Poly(ADP-ribosyl)ation inhibition: a promising approach for ccRCC therapy”, the authors Karpova et al. investigate the effects of reversing hyper poly-ADP-ribosylation (pADPr) apparent in clear cell renal cell carcinomas (ccRCC). The authors show that a number of RCC cell lines and patient samples are enriched in pADPr and argue that metastatic ccRCC samples in particular are the most strongly hyper-pADPribosylated. To observe the effects of reversing hyper-pADPr, the authors treated the cells with PARP1 inhibitors and induced overexpression of PARG, showing that both these treatments are toxic to RCC cells. They attribute this toxicity to the transcriptional role of pADPr in upregulating genes essential for unrestricted growth and tumorigenicity, and attempt to show this by performing RNA sequencing on an RCC cell line overexpressing PARG and finding several cellular pathways impacted by this treatment. However, when the same pathways are investigated in RCC cells treated with PARP inhibition, only some of them show the same effects. Karpova et al. conclude that reverting hyper-pADPr in ccRCC is a promising avenue for therapy.
Although the overall aim to identify therapeutic vulnerabilities of ccRCC is both important and interesting, the conclusions put forth by the authors is not fully supported by the data presented. I do not recommend this study to be published in Cancers unless several key controls and experiments are performed and shown to support the authors’ stipulations. My primary concern involves the mechanism of toxicity observed in this study. I summarize the points to be addressed as follows:
Major points:
- Although the precise mechanism of action of PARP inhibitor toxicity is a key outstanding question, overwhelming evidence is present that genome instability is involved (e.g. chromosomal aberrations, synthetic lethality with multiple genome stability factors). Even if the authors disagree with this paradigm, it should be addressed in the text. Moreover, an experiment showing that genome stability is not the toxicity mechanism seen in the PARPi sensitivity of the RCC cells tested will bolster their model of ‘back-to-normal’ intervention.
- The authors argue that reducing pADPr levels back to normal is toxic specifically in RCC cells. Do the doses where PARPi toxicity is observed coincide with the minimum dose required to reduce pADPr to normal? For PARG overexpression, pADPr is only shown for PNX0010 (Figure S3) and not for the other RCC cells tested (Figure S4); it will be useful to see this, especially since 786-O seems to barely overexpress PARG while having the most profound level of survival defect.
- The PARG experiments need a non-RCC control to show that PARG overexpression toxicity is specific to cancer cells, such as the NK677 control in Figure 2.
- The authors repeatedly discuss the role of PARylation on HIF1α- and HIF2α-dependent transcription and suggests that toxicity of ‘back-to-normal’ pADPr intervention would function through this pathway. However, they provide no compelling evidence that this is the case. The only connection in the data is the downregulation of ID1 and ID2 in PNX0010 overexpressing PARG. If the hypoxia pathway connection is to be made, more evidence needs to be presented that it is at all involved in the toxicity observed.
- It is insufficient to present RNA-seq data and attributing the observed phenotypes to up/downregulation of certain pathways without follow-up experiments. Indeed, the opposite effects seen on lipid and cholesterol metabolism mRNA levels in PARG vs PARPi treatment indicates that this pathway isn’t relevant to the phenotypes observed. More experiments establishing that the proposed pathways are important should be conducted.
- In Figure 2, the survival curves for panels A, B, and E seem to be constructed out of single replicates which is insufficient.
Minor points:
- In Figure 1A, the names of cell lines seem to be mislabeled: presumably the authors meant 769-P and 786-O.
- How were the dosages of drugs used in Figure 2C chosen?
- Representative images of the clonogenic assays from Figures 2 and 3 should be shown in the supplementary.
- In the Discussion line 524, the authors state that the “high level of endogenous pADPr appears to be the best predictor of tumor responsiveness to PARP-1 inhibition”. What is the basis for this statement? The data in the manuscript itself is insufficient for this.
- Size markers on the pADPr blots would be useful. Furthermore, Figure 1A seems to be aggressively cropped, cutting out some visible pADPr signal. A less cropped version would be ideal similar to Figure S2.
Round 2
Reviewer 1 Report
Dear authors, I am satisfied with the experiments carried out and the changes made to the article. Its scientific value has increased. Thanks
Author Response
We thanks reviewer for helpful comments.
Reviewer 2 Report
The authors have addressed the majority of my previous comments to my satisfaction. Though I personally feel that the genome stability role of PARPi can be more discussed and explored, I respect that the authors have included the controls that I requested, giving me increased confidence in the validity of their data. Therefore I recommend that the revised manuscript to be published in its present form in Cancers.
Author Response
We thank the reviewer for helpful comments and suggestions.